# Nutritional Quality and Safety of the *Spirulina* Dietary Supplements Sold on the Slovenian Market

**DOI:** 10.3390/foods11060849

**Published:** 2022-03-17

**Authors:** Jasmina Masten Rutar, Marta Jagodic Hudobivnik, Marijan Nečemer, Katarina Vogel Mikuš, Iztok Arčon, Nives Ogrinc

**Affiliations:** 1Department of Environmental Sciences, Jožef Stefan Institute, Jamova 39, 1000 Ljubljana, Slovenia; jasmina.masten@gmail.com (J.M.R.); marta.jagodic@ijs.si (M.J.H.); 2Jožef Stefan International Postgraduate School, Jamova 39, 1000 Ljubljana, Slovenia; 3Department of Low and Medium Energy Physics, Jožef Stefan Institute, Jamova 39, 1000 Ljubljana, Slovenia; marijan.necemer@ijs.si (M.N.); katarina.vogelmikus@bf.uni-lj.si (K.V.M.); iztok.arcon@ung.si (I.A.); 4Biotechnical Faculty, University of Ljubljana, Jamnikarjeva 101, 1000 Ljubljana, Slovenia; 5Laboratory for Quantum Optics, University of Nova Gorica, Vipavska c. 13, 5000 Nova Gorica, Slovenia

**Keywords:** *Spirulina*, microalgae, cyanobacteria, elements, toxic elements, amino acids, fatty acids, authenticity, safety, quality

## Abstract

The microalgae *Spirulina* may be a popular dietary supplement rich in essential nutrients and vitamins, but oversight of the supplement industry, in general, remains limited, and increasing incidents of adulteration, misbranding, and undeclared ingredients together with misleading claims create potential risks. In response, this study characterized the elemental, amino acid and fatty acid content of commercially available *Spirulina* supplements in Slovenia using EDXRF, ICP-MS and GC-MS and compared the results with their nutritional declaration. The gathered data confirm that *Spirulina* supplements are a good source of calcium (0.15 to 29.5% of RDA), phosphorous (3.36–26.7% of RDA), potassium (0.5 to 7.69% of RDA) and selenium (0.01 to 38.6% of RDA) when consumed within recommended amounts. However, although iron contents were relatively high (7.64 to 316% of RDA), the actual bioavailability of iron was much lower since it was mainly present as the ferric cation. This study also confirms that pure *Spirulina* supplements are a good source of essential and non-essential amino acids, and ω-6 but not ω-3 polyunsaturated fatty acids. The presence of additives resulted in significant variation in nutrient content and, in some instances, lower product quality. Moreover, a high proportion (86.7%) of inappropriate declarations regarding the elemental content was observed. Overall, the study conclusions underline the need for a stricter control system for *Spirulina*-based supplements.

## 1. Introduction

The challenges associated with living sustainability, keeping food production costs down while meeting increasing food demand, has meant that sourcing alternative lipid, protein, pigment and polymer sources has become a global trend. In this context, microalgae rich in functional nutrients that positively affect human health is an excellent example of an alternative nutrient source [1,2,3,4,5]. *Spirulina,* or correctly *Arthrospira* spp., is one of the most important microalgal groups currently produced and contains macro- and micronutrients such as high-quality proteins, minerals, vitamins, fatty acids, polysaccharides and other bioactive compounds [6,7,8,9].

*Spirulina* is a multicellular filamentous cyanobacterium (blue-green microalgae) with nitrogen-fixing symbiotic bacteria. Its multicellular cylindrical trichomes are typically arranged along its entire length in a left-handed open helix, and its surface is without covering and smooth. It is a photosynthetic autotroph with phycocyanin as its primary photosynthetic pigment [10,11]. *Spirulina* is an excellent source of iron, calcium and phosphorous, pigments (carotenoids, c-phycocyanin, chlorophyll-a), and vitamins (vitamin E, vitamin B12). It is also a rich source of digestible proteins (up to 70% of its protein content), polysaccharides and lipids and has a well-balanced amino acid profile. It is also regarded as a good source of essential fatty acids, including ω-6 linoleic and γ-linolenic fatty acid, as well as ω-3 fatty acids such as α-linolenic acid, eicosapentaenoic acid (EPA) and docosahexaenoic acid (DHA) [9,12,13]. It is suggested that this high nutritional value could positively influence the treatment of several pathological conditions such as certain cancers, hepatotoxicity, cardiovascular disease and hyperlipidemia, among others [14,15,16,17]. The lack of cellulose in its cell wall and the absence of phytates and oxalates means that nutrient assimilation in the human gut is high, making it popular among consumers [18].

The *Spirulina* and *Spirulina*-based product market is expected to show continued and rapid growth until 2028 with a compound annual growth rate (CAGR) of 18.1%. Most microalgae-based commercial products are produced in Asia or Australia, while European companies account for approximately 5% of the global food/feed microalgae market [4]. The demand for *Spirulina* products is attributed to increasing health awareness and vegetarianism, malnutrition, dietary supplement intake, and the demand for natural colorings. At present, most production is directed towards food, supplements and nutraceuticals, which together account for 75% of the reported uses [4]. *Spirulina* supplements are segmented into powder, the most popular form, as well as flakes, capsules, tablets, frozen *Spirulina* and phycocyanin extract, and are promoted by producers and suppliers as a health food [19].

The cultivation and commercialization, such as processing and packaging methods and transport of *Spirulina,* can change its chemical composition, affecting its nutritional quality and toxicological properties [20]. Studies have shown the presence of cadmium, mercury, arsenic and lead in *Spirulina* products in anomalous quantities due to pesticide or fertilizer use adjacent to *Spirulina* cultivation areas [21]. Such contamination is of concern because dietary supplements are not strictly regulated or inspected to the same extent as other food products and pharmaceuticals. Further increased market demand and high cost and complexity of *Spirulina* culturing encourage the adulteration of these products with inferior cheaper materials/ingredients, e.g., flour and mung bean powder, resulting in economic losses and potentially putting consumers at risk [22,23].

Combatting adulteration, increasing consumer trust and guaranteeing the quality and safety of *Spirulina* products available on the market requires that products are regularly monitored [23,24,25]. This study focused on the characterization of commercial *Spirulina* supplements sold on the Slovenian market, including elemental, toxic elemental, amino acid, and fatty acid composition, and their compliance with their product declaration and identifying possible adulterations. Furthermore, the bioavailability of Fe was estimated for the first time.

## 2. Materials and Methods

### 2.1. Samples

Forty-six *Spirulina* dietary supplements were purchased from different health food stores/supermarkets and online stores in Slovenia (Table 1). The samples were collected over two months. Forty-four supplements (95.7%) were labelled as *Spirulina* spp., while two samples (4.35%) were mixed with other algae (*Chlorella*, *Lithothamnium*) and plant-based nutritional supplements, e.g., wheatgrass and barley grass. Among the *Spirulina*-only samples, 34 samples (73.9%) were labelled as pure *Spirulina*, while the remaining ten (21.7%) also contained additives. The samples were either fresh or in powder, tablet, or capsule form originating from Italy, Portugal, Japan, China, India, Taiwan, and Hawaii.

### 2.2. Sample Preparation

Fresh samples were freeze-dried and ground to obtain a fine powder, the tablets were also finely ground, and the capsules were opened and the contents used for analysis. The samples were stored in plastic containers and kept refrigerated (4 °C) until analysis. Samples were analyzed in duplicate in small batches. All analyses were performed within one to three months after collection.

### 2.3. Macro-Elemental Analysis by X-ray Fluorescence Spectrometry

The macro-elemental composition (Si, P, S, Cl, K, Ca, Ti, Mn, Fe, Zn, Br, Rb and Sr) was determined non-destructively using Energy Dispersive X-Ray Fluorescence Spectrometry (EDXRF). Pellets (0.5–1.0 g) were prepared using a pellet die and a hydraulic press, and the disc radioisotope excitation sources Fe-55 (25 mCi, Eckert & Ziegler, Berlin, Germany) and Cd-109 (20 mCi, Eckert & Ziegler, Berlin, Germany) used for fluorescence excitation. Fluorescence was measured using the EDXRF spectrometer with an XR-100 SDD detector (Amptek, Bedford, MA, USA), a PX5 digital pulse processor (Amptek, Bedford, MA, USA), and a PC-based, multichannel analyzer software package (DPPMCA). In Fe-55 mode, the spectrometer was equipped with a vacuum chamber to measure light elements (Si, P, S, and Cl), and in Cd-109 mode, in the air for K, Ca, Ti, Mn, Fe, Zn, Br, Rb and Sr. The energy resolution was 125 eV at 5.9 keV. X-ray spectra were analyzed using AXIL Spectral Analysis software. Quantification was performed using the Quantitative Analysis of Environmental Samples (QAES) software developed in-house [26,27]. The estimated uncertainty budget of the EDXRF analysis was 11%. The method was validated by analyzing NIST 1547 (peach leaves) and NIST 1573a (tomato leaves).

### 2.4. Iron Speciation by Fe K-Edge X-ray Absorption Near Edge Structure (XANES)

Powdered *Spirulina* samples were pressed into pellets (0.1–0.3 g), fixed on Teflon holders with Fe free scotch tape and mounted on an LN_2_ cooled stage. X-ray absorption spectra were obtained in fluorescence detection mode using an unfocused CLÆSS beamline at the ALBA synchrotron facility (ALBA, Barcelona, Spain). A pair of horizontal and vertical slits allowed the reduction of the beam size on the sample to about 5 mm × 1 mm, illuminating a major part of the pellet. A silicon (Si 111) double crystal monochromator was used with 1 eV resolution at the Fe K-edge (7112 eV). The samples were inserted between the first and the second ionization cell at 45° relative to the beam. An SDD fluorescence detector, positioned at 90° to the beam, was used to measure the intensity of the Fe-Kα fluorescence radiation. The fluorescence spectra were recorded as the ratio of the fluorescence detector signal and the signal of the incident photon beam from the first ionization chamber with an integration time of 4 s/step. The absorption spectra were measured within the interval −150 eV to 350 eV relative to the Fe K-edge. In the XANES region, equidistant energy steps of 0.2 eV were used and 1 eV steps elsewhere. Three replicates were measured to check scan reproducibility and improve the signal-to-noise ratio. No evidence of Fe K-edge shifts in consecutive scans of the samples was observed due to the absorbed dose of ionizing radiation. The monochromator’s exact energy was calibrated against a 5 μm thick Fe metal foil. The first inflection point in the XANES spectrum of Fe metal was at 7112 eV, while the absolute energy reproducibility was ± 0.03 eV or better. The Fe K-edge XANES spectra were analyzed using the IFEFFIT software package ATHENA [28].

The bioavailability of Fe was estimated by measuring the relative amounts of Fe^2+^ (ferrous iron) and Fe^3+^ (ferric iron). For this, we used the linear combination fit (LCF) method as described in [28,29]. The relative amounts of each Fe cation (Fe^2+^ and Fe^3+^) in the sample were determined based on a linear combination fit of the XANES spectrum of the sample, that of the Fe reference compound with known valence states of Fe.

### 2.5. Toxic Trace Element and Se Analysis by Inductively Coupled Plasma-Mass Spectrometry

Inductively coupled plasma-mass spectrometry (ICP-MS) was used to measure Se, As, Cd, Hg and Pb levels in commercial *Spirulina* samples. Powdered samples (0.05–0.1 g) were weighed into Teflon vials, followed by 2 mL of 65% HNO_3_ (Suprapur^®^, Merck, Darmstadt, Germany). Samples S10, S20, S31, S40, S41, S46 and S47 were prepared in duplicate. The samples were then digested in an UltraWave closed vessel microwave digestion system (Milestone, Sorisole (BG), Italy) at 1500 W and 100 bar maximum pressure. The temperature program was as follows: ramped to 240 °C in 20 min, held for 15 min and then cooled to 40 °C. The digests were then quantitatively transferred to 10 mL polyethylene graduated vials and filled to the mark with MilliQ water. The samples were then filtered through hydrophilic syringe filters (Millipore Millex-HV, Merck, Darmstadt, Germany) (0.45 μm) and diluted in a 1:10 ratio. The reference material BCR-414 (plankton trace elements) and blank samples (HNO_3_) were prepared similarly.

The samples were then analyzed using a triple quadrupole instrument ICP-QQQ (Agilent 8800, Santa Clara, CA, USA) in 1:10 dilution for Se, As, Cd and Hg and 1:100 dilution for Pb. A dilution of 1:100 was used for Hg in the reference material. Calibration curves were prepared for Se, As, Cd and Pb using MULTI XVI (ICP Multi-Element Standard Solution XVI CertiPUR^®^, Merck, Darmstadt, Germany) in 5% HNO_3_. The following concentrations were prepared: 0, 0.1, 0.5, 1, 5, 10, 50, 100, 250 and 1000 ng/mL. The Hg calibration curve was prepared using the NIST 3133 reference material (RM) in a 5% HNO_3_ solution at concentrations of 0, 0.1, 0.5, 1, 5 and 10 ng/mL.

### 2.6. Fatty Acid Analysis by Gas Chromatography-Mass Spectrometry Method

The analysis of fatty acids in *Spirulina* samples was determined using Gas Chromatography-Mass Spectrometry (GC-MS).

#### 2.6.1. Fatty Acid Extraction and Esterification

Powdered *Spirulina* samples (150 mg) were weighed directly into screw-cap vials, and 500 μL of dichloromethane and 3 mL of 0.5 M sodium hydroxide in methanol were added for total lipid extraction. Samples were then purged with nitrogen and heated for 10 min at 90 °C. Once cool, 3 mL of BF_3_-MeOH was added to generate the fatty acids methyl esters (FAMEs) and purged with nitrogen. The samples were heated for 10 min at 90 °C. Once cool, the FAMEs were extracted using 1.5 mL of hexane. The hexane phase was transferred directly into a GC vial and stored at −20 °C. All the samples were prepared in triplicate.

#### 2.6.2. FAME Analysis

Analysis was carried out using a 7890B GC and 5977A Series GC/MSD (Agilent, Santa Clara, CA, USA). Separation was achieved on a 30 m × 0.25 mm × 0.25 μm VF-WAXms capillary column (Agilent J&W, Santa Clara, CA, USA). The injection volume was 1 μL, with a split ratio of 10:1. The carrier gas was helium maintained at a 1.5 mL/min constant flow. The injector temperature was 280 °C and the detector temperature 350 °C. The temperature program was as follows: initial column temperature set at 50 °C for 1 min then programmed to 170 °C at 15 °C/min and held for 5 min, then from 170 °C to 200 °C at 3 °C/min, held 5 min, and from 200 °C to 230 °C at 5 °C/min, and held 17 min.

A standard Supelco 37 component FAME Mix in dichloromethane (Bellefonte, PA, USA) was used for identification and quantification. Compounds were identified based on a comparison of retention times with authentic compounds. With each set of samples, blank samples and the FAME Mix standard were analyzed to verify the stability of the analytical system. The results are expressed as the weight percent of an individual fatty acid to the total fatty acid (TFA) content calculated from the peak area using the appropriate correction factors [30].

### 2.7. Amino Acid Analysis by Gas Chromatography-Mass Spectrometry

The amino acid composition was determined using Gas Chromatography-Mass Spectrometry (GC-MS).

#### 2.7.1. Liquid Phase Hydrolysis

The total amino acid extraction was based on hot protein hydrolysis and simultaneous free amino acid solubilization. Before the analysis, the hydrolysis micro-reaction vessels (5 mL, heavy-wall borosilicate glass, 20 mm × 65 mm, screw top, with a solid phenolic cap) were cleaned by pyrolysis at 500 °C for 6 h and left to cool overnight. The hydrolyzing agent (6N HCl) was prepared fresh from a 30% hydrochloric acid solution. The samples (15 mg) were weighed directly into the reaction vials and the hydrolyzing agent (1 mL) containing 4% thioglycolic acid, which acts as a reducing agent to prevent amino acid oxidation. After, 1% phenol was added to prevent halogenation of the tyrosine. Oxidation was prevented by purging the samples with N_2_ (5 min). The vials were then sealed and heated at 110 °C for 24 h.

#### 2.7.2. Amino Acid Derivatization

For derivatization, a commercial EZ:faast Amino Acid Hydrolysate kit (Phenomenex, Torrance, CA, USA) was used. The procedure was as follows: 355 μL of the Na_2_CO_3_ solution was added to the hydrolysate sample (100 μL) to obtain a pH of 2–2.5. To this was added 20 μL of the norvaline internal standard solution (0.2 mM) and 100 μL 10% n-propanol. The samples were then extracted using solid-phase extraction and the amino acids eluted with 200 μL freshly prepared eluting medium (sodium hydroxide:n-propanol in 3-picoline, 3:2, v/v). Further, the amino acids were derivatized in a mixture of chloroform and propyl chloroformate (50 μL). The amino acids were then extracted into the organic chloroform layer by repeated emulsification and allowing the reactions to proceed for 1 min in between vortexing. Iso-octane (100 μL) was then added, and the mixture was emulsified for an additional 5 s and allowed to react for 1 min. The organic layer was then transferred into a GC vial and reduced to dryness (N_2_). The amino acid derivatives were then reconstituted in a solution (100 μL) of iso-octane:chloroform (80:20, v/v). All samples were prepared in duplicate.

#### 2.7.3. Gas Chromatography-Mass Spectrometry Method for Amino Acid Analysis

Amino acid analysis was performed using a 7890B GC and 5977A Series GC/MSD (Agilent, Santa Clara, CA, USA). Separation was achieved on a 10 m × 0.25 mm × 0.15 mm ZB-AAA GC column provided in the EZ:faast kit together with a FocusLiner^®^. The split ratio was 15:1, and the injector temperature was 250 °C. The injection volume was 1.5 μL. Helium was used as the carrier gas at a 1.5 mL/min flow rate. The temperature program was 110 °C to 320 °C at 30 °C/min. The detector temperature was set to 310 °C. Calibration curves were prepared for individual amino acids at concentrations of 50, 100 and 200 nmol/mL using the standard mixture (SD) provided. The amino acid standard mixture consisted of 200 nmoles/mL of each amino acid: alanine (ALA), glutamic acid (GLU), hydroxylysine (HLY), leucine (LEU), phenylalanine (PHE), threonine (THR), valine (VAL), aspartic acid (ASP), glycine (GLY), hydroxyproline (HYP), lysine (LYS), proline (PRO), tryptophan (TRP), cystine (C-C), histidine (HIS), isoleucine (ILE), methionine (Met), serine (SER) and tyrosine (TYR). Each calibrant was prepared in triplicate. From then on, the SD solutions were treated following the same procedure as the samples. Individual amino acids were identified by comparing peak retention times with known amino acids in the standard. The amino acid content results are expressed in mg/g of sample dry weight (dwt).

### 2.8. Statistical Analysis

Statistical analysis was performed using XLSTAT software (Addinsoft, Long Island, NY, USA, 2019). First, basic statistical methods were used for data analysis (median and quartiles, minimum, maximum, average). Principal component analysis (PCA) was applied further to identify characteristic parameters to discriminate samples based on their macro and trace-elemental composition, amino acid and fatty acid composition. The results are presented as biplots, simultaneous variables and as PCA plots.

## 3. Results and Discussion

### 3.1. Elemental Composition

The results are presented in Table 2. The elemental content in the supplements was as follows: Se < Rb < Br < Ti < Zn < Sr < Mn < Fe < Ca < Cl < Si < S < P < K. We specifically focused on elements Fe, Ca, K, Se and P since *Spirulina* is promoted as a rich source of these elements [13,15,31].

Although the recommended daily intake (RDI) varies, the majority of the producers recommend 3–10 g of *Spirulina* supplement intake per day, regardless of the product form (powder, capsule or tablet). As well as the RDI, the data are further evaluated using Dietary Reference Values (DRV) such as Population Reference Intake (PRI), which represents the level of nutrient intake adequate for all people in a population group, or Adequate Intake (AI), which is used when a PRI cannot be determined. Where possible, the elemental composition was also evaluated in terms of Tolerable Upper Intake Level (UL), i.e., the maximum daily intake of a nutrient from all sources unlikely to pose adverse health effects on humans [32]. The data are presented as the median value and interquartile range (IR, in parentheses).

Given the recommended daily dose, the minimal intake of Ca is 1.38 to 4.60 mg Ca/day, the maximal Ca intake is 84.0 to 280 mg Ca/day, and a median value of 4.38 (2.92–7.8)–14.6 (9.72–26.0) mg Ca/day. The PRI for Ca is 950 mg/day [33]. Therefore, the values are within the recommended PRI values of 0.15% to 29.5%. This variability can be attributed to the differences in supplement formulation and the presence of additives. For example, values were found in S19, which contains calcium carbonate, and S1, which contains calcium carbonate in the form of edible scallop shell powder [34]. The lowest values were in S41, S42 and S46. The highest amount of Ca was in S18, a mixed sample containing *Spirulina*, *Chlorella* and *Lithothamnium* algae (191–635 mg/day for 3 g to 10 g of supplement/day, respectively). This value is likely due to *Lithothamnium calcareum* (*L. calcareum*), a seaweed that crystallizes calcium carbonate in its cell walls [35]. However, the UL of 2500 mg/day is unlikely to be exceeded by including *Spirulina* supplements in the diet [36].

The minimal and maximal P intake based on RDI was 18.5–61.6 and 44.1–147 mg/day, respectively. The median value was 32.7 (30.3–35.9)–109 (101–120) mg P/day. The P content represents 3.36–26.7% of the AI (550 mg/day) [33]. The lowest value among all samples was in S9, a mixed sample containing wheatgrass, barley grass, *Chlorella* and *Spirulina*, i.e., 15.2 (for 3 g of supplement/day)–50.6 mg/day (for 10 g of supplement/day). This most likely results from the lower amount of *Spirulina* and *Chlorella* algae in the supplement, especially since *Spirulina* and *Chlorella* typically contain much higher phosphorous content than cereal grasses [13,15,37]. Like Ca, if the *Spirulina* supplement consumption remains within the RDI, the UL determined for P (3000 mg/day) [36] is unlikely to be exceeded.

The minimal intake of K varies from 17.5 to 58.3 mg/day and maximal from 80.7 to 269 mg/day, with a medium intake of 45.6 (42.8–50.0)–152 (143–167) mg/day. This amount would account for 0.5–7.69% of the AI value (3500 mg/day) for an adult [33]. Three of the highest K values were in S5, S44 and S46, where S44 and S46 originate from Italy. The lowest values were measured in S37 and S22, which contained additives and did not declare an origin. No UL was set for K consumption.

The daily intake of Fe (mg Fe/day) was from 0.84–2.81 (minimal value) and 10.4–34.8 (maximal value) with a median of 2.07 (1.47–3.40)–6.89 (4.90–11.3). Such amounts account for between 5.25 to 218% of the PRI for females (16 mg/day) and 7.64 to 316% for males (11 mg/day) [33]. The high deviation observed among the samples is due to the high amount of Fe in S4 and S26 from Hawaii and S2 with no declared origin. High Fe values likely result from a high concentration of Fe in the growth medium. The Fe content in the *Spirulina* microalgae has been proven to reflect that in the growth medium [38,39].

Iron bioavailability in *Spirulina* was determined by analyzing the relative ferrous and ferric iron amounts using XANES analysis. The results are presented in Table 3.

The results show that most iron (82–92%) is present as Fe^3+^, which means that the bioavailability of Fe from the supplements is low, as only a small amount of iron is available in a more bioavailable ferrous form, Fe^2+^ [40,41,42]. This finding also means that promoting *Spirulina* as a rich source of dietary iron should be reconsidered.

The minimal and maximal Se intake was 0.01 to 0.04 and 8.10 to 27.0 μg/day, respectively, with the median being 0.30 (0.20–0.80)–1.01 (0.66–2.67) μg Se/day. The amounts of Se also varied significantly, but S14, S15 and S32 stand out due to their high Se value. The former two are from Taiwan, and the latter is from China. High Se values in these samples are believed to be due to higher amounts of Se in the growth medium. However, Se from these types of samples has been shown to have a lower bioavailability than classical sources such as selenomethionine and inorganic Se salts. In addition, Se from *Spirulina* is metabolized differently due to its chemical form [43]. The lowest Se values were determined in pure *Spirulina* samples from Italy (S44 and S46). The Se accounts for 0.01% to 38.6% of the recommended AI of 70 μg/day [33]. The UL for Se is 300 μg/day [36] but is unlikely to be exceeded by adding daily *Spirulina* supplements to a regular diet.

Determined values of all elements are similar to those previously reported in the literature [13,15,20,44]. However, the significant variability observed in the amounts of certain elements is likely related to the growth medium used and intentional enrichment. It has been shown that micronutrients in the growth medium significantly improve uptake and accumulation of the macro- and micronutrients [39,43,45]. The growth medium pH can also affect *Spirulina* mineral assimilation, i.e., metal ion assimilation increases at higher pH [46]. According to the data, *Spirulina* food supplements are a good source of iron, calcium and phosphorous and can provide substantial amounts of potassium and selenium. However, the intake of specific nutrients is product dependent and depends on the amount of supplement consumed, since the recommended daily consumption values differ among producers.

The high iron content in *Spirulina* supplements is significant for those who consume, for example, fewer foods of animal origin and therefore have a lower iron intake in their diet. In addition to containing high amounts of iron, *Spirulina* also does not contain phytates or oxalates that would cause iron chelation, making *Spirulina* iron highly available for absorption in the human intestine [15,47]. However, as this study has shown, more research is needed to assess the actual iron bioavailability from *Spirulina* due to the predominance of the ferric (Fe^3+^) iron form.

#### Compliance with Their Nutrient Declaration

Measured elemental values were compared to the values declared on the products. Table 4 lists the 15 products that provided information on the content of Fe, Mn, Ca, Zn and P together with the degree of deviation (%) from the declared values. Iron had the most declarations (32.6% of the products), followed by Mn (13.0%), Ca and Zn (8.70%), P (4.3%) and finally K and Se (2.17%). The maximum permissible deviation of mineral content in food supplements is from −20% to 45% [48]. An excessive negative deviation from the declared Fe content was found in S5, S24 and S28 and a positive deviation in S4, S8, S10, S13, S14, S15, S17 and S36. The Mn content deviated positively in S4 and negatively in S12, S13 and S17, while S28 had an insufficient Ca content. The Zn content was low in S17, S23 and S24, while the P content was high in S12. In the case of K, the measured content in S17 was within the declared limits (+10.7%), while Se was much lower (−95.8%) than the declared value. However, several producers state that mineral levels can deviate due to seasonal fluctuations. In addition, even though some values are high, they remain below the UL and are unlikely to pose a risk to human health. However, the proportion of inappropriate declarations (86.7%) is a cause for concern and could undermine consumer confidence, and supports the need for regular monitoring and improved quality control.

### 3.2. Toxic Trace Element Content

The content of Cd, Hg and Pb was evaluated according to the maximum allowed European Commission levels [49]. In contrast, the As content was evaluated according to benchmark dose lower confidence limit (BMDL_01_) for cancers of the lung, skin and bladder, and skin lesions determined by The Scientific Panel on Contaminants in the Food Chain (CONTAM) [50]. Toxic trace element content and their maximum allowed values are presented in Table 5.

The maximum measured values of Cd and Pb were 226 μg/kg dwt and 1320 μg/kg dwt, respectively, and did not exceed the maximum allowed value in food supplements. The daily BMDL_01_ values were calculated based on the average male (87 kg) and female (68 kg) body weight in the Slovenian population [51]. The BMDL_01_ values for As range between 0.3 and 8 μg/kg b.w. per day. Based on an RDI of 3–10 g/day, the maximum daily intake from consuming *Spirulina* was 8.11–27.0 μg/day and did not exceed the upper BMDL_01_ value for all of the supplements tested. In the case of Hg, S39 exceeded the maximum value of 100 μg/kg dwt by 1% (101 μg/kg dwt). All others were below the maximum allowed Hg value in food supplements. The samples’ median value (interquartile range) was 5.57 (3.57–7.63) μg/kg dwt. The lowest values of As, Cd and Pb were found in S46 produced in Italy, while the lowest Hg value was measured in S18, a mixed sample containing *Spirulina*, *Chlorella* and *Lithothamnium* algae. The measured values of elements Cd, Hg and As are comparable to literature values [6,21,52,53]. Pb values are also consistent with those determined by Al-Homaidan [53] but are lower than in commercial samples tested by Hsu et al. and Campanella et al., which ranged between 5600–15,200 μg/kg dwt, and 30.8% of the *Spirulina* samples tested by Rzymski et al., which ranged from 3500 μg/kg dwt to 5000 μg/kg dwt. The authors attributed the high values to local contamination, greater propensity toward Pb assimilation by microalgae and natural background levels [6,52,54,55].

Based on the data from the present study, *Spirulina* supplements do not contribute significantly to the intake of toxic trace elements and do not pose a serious risk. Nevertheless, *Spirulina* is an effective accumulator of trace elements which is an advantage when it accumulates elements essential for human health but a disadvantage for toxic elements [21]. *Spirulina* is also a potential source of Pb, Hg, Cd, and As in open production systems, where pedoclimatic conditions and agricultural practices could contribute to their presence in higher values and again supports the need for regular monitoring [56,57,58]. In contrast, this is not an issue when grown in controlled (closed) environments [21].

### 3.3. Amino Acid Content

Amino acids asparagine (ASN) and glutamine (GLN) were quantitatively converted to aspartate (ASP) and glutamate (GLU) during acid hydrolysis. Tryptophan (TRP) was lost during acid hydrolysis, and arginine (ARG) and cysteine (CYS) were not included in the GC-MS EZ:faast kit due to their thermal instability [59,60]. Again, the results (Table 6) are presented according to the RDI of 3–10 g. The measured values are compared to the daily recommended amounts [61] for the average male (87 kg) and female (68 kg) in Slovenia [51].

The amino acid values determined in this study are presented in Appendix A and are within the literature values, except for VAL and HIS, where the measured values were higher [6,62,63]. The VAL content ranged from 50.8–123 mg/g dwt in this study, while the previously reported values range from 35.8–60 mg/g dwt. The HIS content varied from 16.3–22.2 mg/g dwt, compared to 6.00–11.9 mg/g dwt reported in the literature (Figure 1; Appendix A). The lowest content of all amino acids except MET was determined in S22, closely followed by S19 and S37. These samples contain *Spirulina* and excipients, especially S22 and S19, which have the most declared excipients among all the samples. It is known that a high excipient content affects the amino acid content due to the reduced amount of *Spirulina* [6]. Valine, ILE and HIS were present in the lowest amounts in all samples. The sample with the second-lowest content of all other analyzed amino acids was S9—a mixed sample of wheatgrass, barley grass, *Spirulina* and *Chlorella*. This result could also be due to the low microalgae content in the supplement, as cereal grasses are not as rich in protein as *Spirulina* and *Chlorella* [64,65]. These results suggest that, combined with elemental composition, the amino acid content could distinguish authentic samples from adulterated ones. The highest values of ALA, GLY, VAL, LEU, ILE, THR, SER, ASP and TYR were found in S36 and S40, PRO and LYS were highest in S6, PHE in S8, HIS in S28, MET in S39 and GLU in S46—all were declared as pure *Spirulina* products.

In addition to product composition, other factors affecting the amino acid content have been reported. For example, cells grown under stress conditions, including salinity stress, have a lower capacity for protein synthesis, which is seen in the lower protein content found in biomass grown in salinated water [66,67]. Differences also occur due to different cultivation times, light intensities, temperatures, and the growth medium’s nutrient composition. For example, *Spirulina* grown in a urea growth medium has a higher amino acid content than others [62]. The drying processes used can also affect the amount of protein and, consequently, amino acid content. The highest protein losses are connected to convective and infrared drying in spreading cylinders, while freeze-drying gives the highest protein recoveries. In addition, thin layer drying results in higher protein recoveries than cylinder drying [68]. Drying at low temperatures (40–50 °C) does not affect the products’ nutritional quality compared to the fresh *Spirulina* [69].

This study shows that the *Spirulina* supplements contained all the essential and non-essential amino acids measured. Coverage of daily requirements for adults [61] based on median values (%) was: TYR (males (M): 4.79–16.0, females (F): 6.13–20.4), LEU (M: 4.99–16.6, F: 6.39–21.3), PHE (M: 4.99–16.6, F: 6.38–21.3), HIS (M: 6.75–22.5, F: 8.63–28.8), MET (M: 7.06–23.5, F: 9.03–30.1), LYS (M: 8.85–29.5, F: 11.3–37.8), THR (M: 11.4–38.1, F: 14.6–48.7), ILE (M: 12.6–42.1, F: 16.2–53.9) and VAL (M: 13.6–45.2, F: 17.4–57.8). According to these results, the tested *Spirulina* supplements are a good source of essential and non-essential amino acids, as shown by other authors analyzing commercial *Spirulina* products [9,63,66]. However, despite that S22, S19 and S37 stand out because of their low amino acid content, overall, *Spirulina* food supplements would be reasonable when choosing high amino acid content products.

### 3.4. Fatty Acid Content

The amounts of individual fatty acids (FA) are presented in Table 7 and Appendix A. The results in Table 7 are presented as the median value (interquartile range in parentheses) of weight percent of individual fatty acids of total fatty acids (TFA). The most abundant FA was palmitic acid (16:0) followed by linoleic (C18:2 ω-6c), γ-linolenic (C18:3 ω-6), palmitoleic (C16:1 ω-7), stearic (C18:0) and oleic/elaidic acid (C18:1 ω-9c/9t). The amounts of fatty acids agree with published data [6,70], although levels of γ-linolenic, linoleic, palmitoleic and stearic acid vary [71,72,73,74]. These differences could be due to higher or lower content of saturated/unsaturated FA in the algal cells, which depends on metabolic needs, such as membrane fluidity, which depends on the growth conditions [75]. It is known that certain microalgae, including *Spirulina*, regulate their lipid composition at low temperatures to achieve better membrane fluidity, which results in increased levels of unsaturated FA.

High temperatures during growth also favor the formation of saturated FA [70,76,77], and the formation of certain fatty acids is more intensive at specific temperatures for certain strains, i.e., the highest γ-linolenic FA production in *Spirulina maxima* is between 35 and 40 °C. In contrast, in *Spirulina platensis*, the production of γ-linolenic is higher at 30 °C [78]. Culture age and growth medium salinity are also important, and prolonged exposure to high salinity stress results in higher values of γ-linolenic acid in different *Spirulina* strains [72]. As with amino acids, the drying process can affect lipid yield and total fatty acid content. Namely, the maximum monounsaturated and polyunsaturated FA levels are obtained in the freeze-dried samples, while the sun-dried samples contain higher amounts of saturated acids. In addition, freeze drying provides better nutrient preservation than other dehydration methods such as sun drying, oven drying or spray drying [79].

In most samples, only small amounts or no ω-3 fatty acids were found, making commercial *Spirulina* samples a good source of ω-6 (especially LA and GLA), but not ω-3 fatty acids, which was also confirmed by other studies [6,70,74,80].

Like elemental and amino acid composition, certain samples stood out regarding TFA content (Figure 2). According to median and associated IR values, anomalously high levels of C15:0 (0.22% of TFA), C16:2 ω-6 (9.71% of TFA), C16:2 ω-4 (9.38% of TFA), C16:3 ω-3 (9.58% of TFA) and C18:3 ω-3 (18.4% of TFA) were found in S6. Additionally, in the same sample, anomalously low levels of C16:0 (18.4% of TFA), C17:1 (< LOD of TFA) and C18:3 ω-6 (< LOD of TFA) were observed, suggesting possible adulteration. S19 contained unusually high levels of C18:0 (25.8% of TFA), and low values of C17:1 (< LOD of TFA), C18:2 ω-6c (12.1% of TFA) and C18:3 ω-6c (9.72% of TFA); likewise, S37 had lower amounts of C18:2 ω-6c (11.1% of TFA) and a marked deviation in its C22:0 content (30.2% of TFA). The presence of various excipients could explain these differences. However, for S37, there remains the possibility of adulteration since it contains high levels of C22:0 FA, which was otherwise detected only in S6 (0.27% of TFA, possibly adulterated), and in the mixed sample S9 (0.59% of TFA), which contains wheatgrass, barley grass, *Spirulina* and *Chlorella*. Sample S9 also stands out due to its high content of C14:0 (0.84% of TFA), C18:3 ω-3 (22.8% of TFA), C16:2 ω-6 (5.52% of TFA), and C16:3 ω-3 (5.27% of TFA) and low content of C16:1 ω-7 (2.75% of TFA), C16:0 (24.6% of TFA) and C18:3 ω-6 (5.69% of TFA).

In addition, S18 differed from supplements containing only *Spirulina*. This sample contains high amounts of C16:2 ω-6 (14.6% of TFA), C18:2 ω-6c (34.4% of TFA) and C18:3 ω-3 (2.52% of TFA). In contrast, the amounts of C16:0 (28.3% of TFA), C18:0 (0.88% of TFA) and C18:3 ω-6 (8.84% of TFA) were exceptionally low. Higher levels of C18:3 ω-3 (ALA, α-linolenic acid) in mixed samples likely result from the presence of *Chlorella*, which is known to contain large amounts of this FA. Alternatively, low GLA and palmitic acid levels point to a low *Spirulina* content in S9 and S18, as their high content is typical for pure *Spirulina* products [74]. The fatty acid distribution (% of TFA) in the analyzed samples is presented in Figure 2.

The data make it possible to distinguish between mixed samples and samples containing pure *Spirulina* based on the fatty acid composition. Such knowledge could help separate authentic from adulterated samples. In addition, the *Spirulina* supplements tested proved to be a good source of ω-6, but not ω-3 polyunsaturated fatty acids. However, since the FA content varied in samples, it would be sensible to choose pure *Spirulina* products for consumption with no added excipients from a trusted producer to guarantee optimal FA composition.

### 3.5. Principal Component Analysis (PCA) of Spirulina Samples from Slovenian Market Analysis Results

The data set of 46 *Spirulina* samples and 43 analyzed parameters (macro- and trace-elemental, amino acid and fatty acid composition data) was analyzed by PCA to identify the trends and examine the distribution of variables in the investigated samples (Figure 3 and Figure 4). PC1 explained 33.0% and PC2 13.5% of the total variance. Four groups of samples can be identified (Figure 3), each represented by different variables. Most parameter vectors are directed towards the upper right quadrant of Figure 4. The positive trend is due to an increased amino acid composition as well as C16:0, C16:1 ω-7, C18:2 ω-6c and C18:3 ω-6 fatty acids. The blue group (Figure 3) has the highest content of these compounds, followed by the yellow group, while the red group has the lowest amount. Finally, ungrouped samples (S6, S9, S18, S19, S22 and S37), which have the lowest content of selected amino acids and fatty acids, are located on the left-hand side of the graph. A similar positive trend for elements Cl, Ti, Fe, Zn, Mn, Rb and Br and fatty acids C14:1 and C18:1 ω-9c/C18:1 ω-9t is observed in the upper part of the second PCA graph (Figure 4). The samples from Hawaii (orange group) contained the highest amounts of these elements and fatty acids. Interestingly, the undeclared sample S2 falls in the same group, indicating that this *Spirulina* supplement might originate from Hawaii. Alternatively, ungrouped samples to the left of the vertical line of Figure 3 (S6, S9, S18, S19, S22 and S37) show a positive Ca, Sr and Cd and C14:0, C18:0 and C16:2 ω-4 trend (Figure 4).

Samples S9 and S18 were expected to stand out since they contain other plant or algae material, and their parameters were expected to differ from the samples containing pure *Spirulina*. In addition, S19 and S22 contain various excipients, affecting the nutrient composition. In contrast, close inspection of the data for S6 and S37 suggests possible adulteration since S6 is declared as pure *Spirulina* and S37 contains excipients, which did not affect our results. Overall statistical evaluation of the results supports our previous observations.

## 4. Conclusions

This study finds that when consumed in recommended amounts, the analyzed *Spirulina* supplements are a good source of calcium, phosphorous, potassium, and selenium, while toxic trace elements do not represent a serious health risk. They are also a good source of essential and non-essential amino acids as well as ω-6 polyunsaturated fatty acids. The study data also show that *Spirulina* contains low amounts of ω-3 polyunsaturated fatty acids, and although they contain high amounts of Fe, since it is mainly present as Fe^3+^, the Fe is less bioavailable.

Therefore, a well-thought-out selection of *Spirulina* supplements would be advised and choosing pure *Spirulina* supplements is advisable, especially since pure *Spirulina* samples have a higher amino acid, γ-linolenic and linoleic fatty acid, P and Se content. Additionally, as different *Spirulina* products might have a different nutrient composition, the products should be chosen according to the specific nutrient needs of the individual. Supplements from Hawaii are rich in Fe, Zn, Mn, Cl, Ti, Rb and Br.

Notably, a high proportion (86.7%) of inappropriate declarations was found among the analyzed samples regarding the content of Fe, Mn, Ca, Zn, P, K and Se, which is a cause for concern. Deviations of more than 45% over the declared value could pose a risk to human health through excessive elemental intake. Such deviations can also undermine consumer confidence. Fortunately, in this case, UL levels were not exceeded.

This study also showed how adding algal or plant material supplements alters the elemental, amino acid and fatty acid composition. Such data could distinguish mixed products from those containing only *Spirulina* and therefore be valuable in authenticity studies. Multivariate analysis was able to discern these products from those containing only *Spirulina*. In addition, the amino acid, fatty acid and mineral composition data suggest that at least two samples were adulterated, since such differences in composition compared to pure *Spirulina* could not be explained by the addition of excipients.

Overall, *Spirulina* is a good source of nutrients. However, in the case of dietary supplements, regular monitoring and inspection are advised to identify adulterations and deviations from the declared content. In this way, potential hazards for consumer health can be avoided.

## Figures and Tables

**Figure 1 foods-11-00849-f001:**
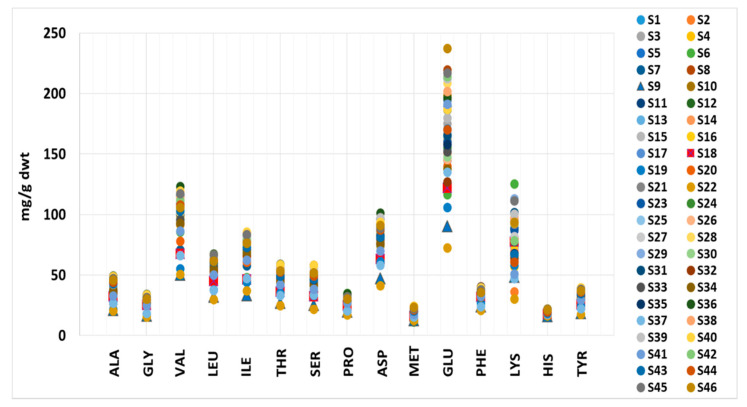
The amino acid content range in *Spirulina* supplements.

**Figure 2 foods-11-00849-f002:**
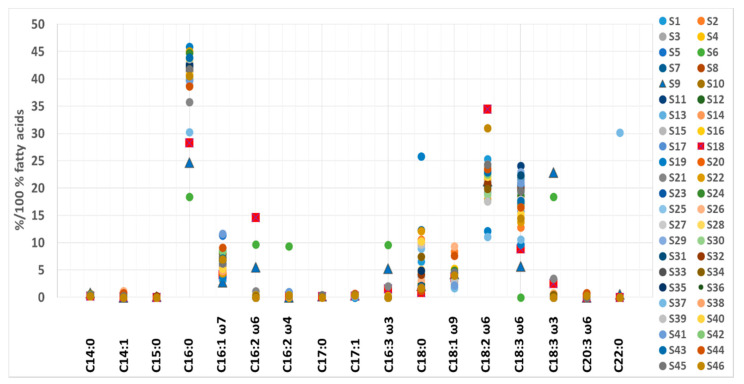
The fatty acid content range in *Spirulina*-based supplements.

**Figure 3 foods-11-00849-f003:**
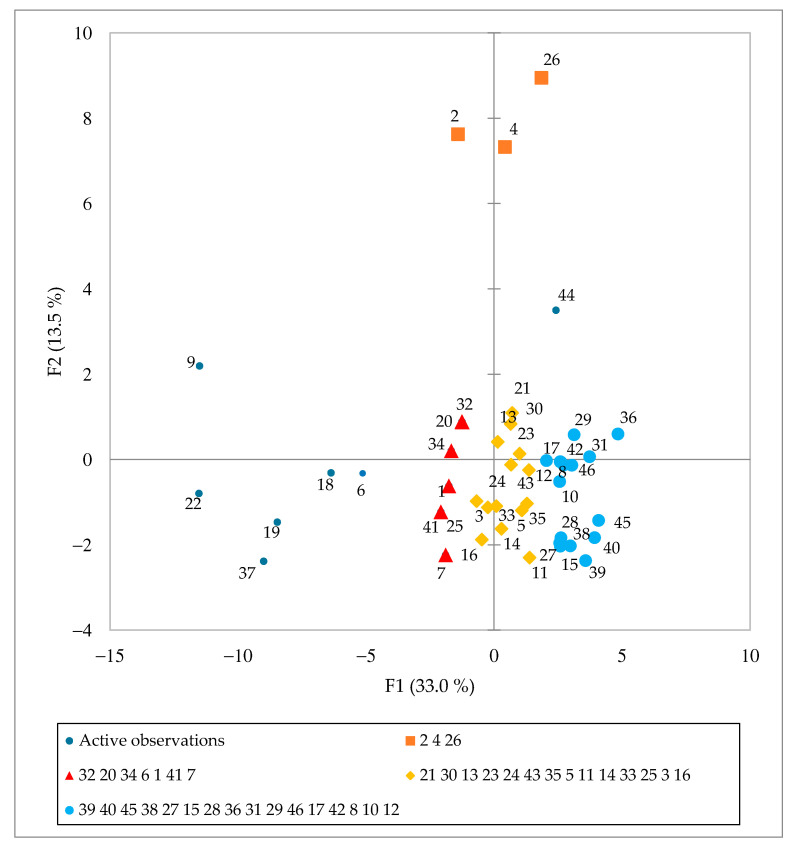
Principal Component Analysis score plot of *Spirulina* dietary supplements (*n* = 47) available in Slovenia.

**Figure 4 foods-11-00849-f004:**
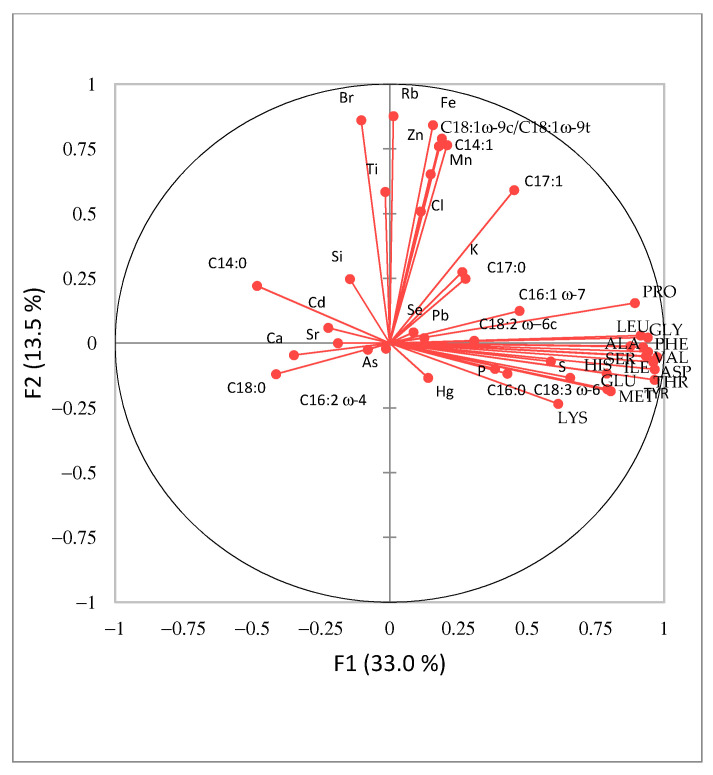
Principle Component Analysis variable loading plot for *Spirulina* dietary supplements (*n* = 47) available in Slovenia.

**Table 1 foods-11-00849-t001:** List of *Spirulina* supplements purchased from the Slovenian market, including their origin, product content and form as declared on the product label. The samples are clustered according to the declared country of origin.

Sample	Form	Declared Origin	Declared Product Content
S1	Tablets	Japan	*Spirulina*, edible scallop shell powder, edible refined processing fat
S4	Tablets	Hawaii	*Spirulina pacifica*, silicon dioxide, chicory inulin, magnesium stearate
S26	Tablets	Hawaii	*Spirulina pacifica*, silicon dioxide
S7	Tablets	India	*Spirulina platensis*
S19	Tablets	India	*Spirulina platensis*, calcium carbonate, micro-crystalline cellulose, stearic acid, croscarmellose sodium, silica
S34	Tablets	India	*Spirulina platensis*
S38	Tablets	India	*Spirulina platensis*
S9	Powder	Mongolia-China	Wheatgrass, barley grass, *Spirulina*, *Chlorella*
S8	Tablets	China	*Spirulina platensis*
S10	Powder	China	*Spirulina*
S11	Powder	China	*Spirulina platensis*
S12	Powder	China	*Spirulina*
S13	Tablets	China	*Spirulina*
S17	Powder	China	*Spirulina platensis*
S23	Tablets	China	*Spirulina*
S24	Powder	China	*Spirulina*
S25	Tablets	China	*Spirulina*
S27	Powder	China	*Spirulina*
S31	Powder	China	*Spirulina platensis*
S32	Tablets	China	*Spirulina*
S33	Powder	China	*Spirulina*
S41	Powder	China	*Spirulina platensis*
S43	Powder	China	*Spirulina*
S3	Powder	Outside EU	*Spirulina platensis*
S5	Powder	Outside EU	*Spirulina*
S6	Powder	Outside EU	*Spirulina platensis*
S16	Powder	Outside EU	*Spirulina*
S35	Tablets	Outside EU	*Spirulina*
S36	Powder	Outside EU	*Spirulina*
S14	Tablets	Taiwan	*Spirulina platensis*
S15	Powder	Taiwan	*Spirulina platensis*
S40	Tablets	Taiwan	*Spirulina platensis*
S29	Tablets	Portugal	*Spirulina platensis*, silicon dioxide, magnesium stearate
S30	Tablets	Portugal	*Spirulina platensis*, silicon dioxide, magnesium stearate
S44	Flakes	Italy	*Spirulina platensis*
S46	Fresh	Italy	*Spirulina platensis*
S18	Capsules	EU	*Spirulina, Chlorella, Lithothamnium*
S2	Capsules	NS ^1^	*Spirulina pacifica*, magnesium stearate
S20	Tablets	NS	*Spirulina*, silicon dioxide, magnesium stearate
S21	Tablets	NS	*Spirulina*
S22	Tablets	NS	*Spirulina platensis*, maltodextrin, silicon dioxide, magnesium stearate, hydroxypropyl methyl cellulose
S28	Tablets	NS	*Spirulina*
S37	Tablets	NS	*Spirulina maxima*, corn maltodextrin, magnesium stearate
S39	Tablets	NS	*Spirulina*
S42	Capsules	NS	*Spirulina*
S45	Powder	NS ^1^	*Spirulina*

^1^ NS—Not Specified.

**Table 2 foods-11-00849-t002:** Macro-element composition of *Spirulina* supplements available on the Slovenian market.

S. No. ^1^	Si (g/kg)	P (g/kg)	S (g/kg)	Cl (g/kg)	K (g/kg)	Ca (g/kg)	Ti (mg/kg)	Mn (mg/kg)	Fe (g/kg)	Zn (mg/kg)	Se (ug/g)	Br (mg/kg)	Rb (mg/kg)	Sr (mg/kg)
S1	1.57	10.5	7.53	0.66	16.1	8.18	18.2	43.4	0.81	14.0	0.06	1.29	1.55	37.9
S2	7.69	11.4	7.53	3.07	14.8	1.32	56.9	159	3.29	43.6	0.59	11.2	7.47	10.0
S3	1.16	10.6	8.64	3.18	15.4	0.82	5.45	29.4	0.49	8.33	0.12	1.91	1.67	17.8
S4	13.5	10.0	7.57	5.77	17.1	2.20	46.5	128	3.48	52.7	0.42	16.5	11.9	24.3
S5	1.34	12.4	9.04	3.78	20.8	0.74	8.77	22.1	0.48	14.6	0.06	1.77	1.23	25.1
S6	21.7	12.7	6.47	0.09	11.0	1.28	9.31	54.9	0.90	13.9	0.02	0.50	4.12	10.2
S7	10.7	11.8	7.77	0.60	14.3	5.10	3.96	32.8	0.37	10.4	0.03	0.47	0.81	22.1
S8	16.0	11.5	7.53	1.03	14.7	2.04	60.6	35.0	1.39	16.0	0.40	0.91	0.91	27.3
S9	5.21	5.06	3.14	2.61	19.7	2.83	10.4	43.0	0.44	19.1	0.03	10.8	3.75	12.7
S10	2.34	14.1	9.38	2.11	18.5	3.09	35.5	36.6	1.68	18.5	0.37	1.34	1.58	34.7
S11	1.42	12.9	8.29	0.48	15.2	1.20	4.42	26.5	0.57	11.1	0.10	1.19	1.07	28.2
S12	1.63	13.9	8.42	1.77	16.8	5.24	12.3	38.2	1.38	33.0	0.15	1.39	1.48	31.8
S13	16.6	12.6	7.79	1.97	15.5	5.39	14.8	34.4	1.79	33.6	0.14	1.47	0.92	31.1
S14	7.94	12.2	7.54	0.19	13.6	1.00	9.91	34.9	0.69	16.5	2.70	0.57	2.23	7.41
S15	1.43	11.9	7.32	0.21	13.7	0.89	5.07	33.3	0.66	15.7	2.63	0.48	1.60	6.82
S16	1.59	14.7	7.50	0.52	14.3	2.78	6.08	36.3	0.65	17.5	0.07	0.91	0.50	12.7
S17	1.61	13.5	8.64	2.21	16.6	5.34	12.3	30.9	1.74	34.9	0.13	1.88	1.20	32.3
S18	2.74	12.6	6.17	0.60	8.63	63.5	43.1	47.1	0.75	11.1	0.11	9.11	2.55	478
S19	19.4	12.2	9.29	2.55	18.4	28.0	15.9	30.3	0.56	9.74	0.10	2.26	1.24	27.6
S20	16.8	12.1	8.39	3.04	16.3	2.43	47.5	29.1	1.13	23.0	0.92	2.00	1.91	32.2
S21	14.7	9.77	7.30	1.94	16.2	1.35	28.2	150	0.69	22.4	0.07	3.09	2.97	9.66
S22	1.85	6.82	3.60	0.55	7.40	1.37	11.1	21.1	0.39	7.69	0.07	1.24	1.06	8.00
S23	15.1	11.2	7.12	1.12	14.2	2.03	65.8	28.3	1.39	14.0	0.44	0.88	0.78	28.0
S24	1.40	9.27	6.05	0.87	12.5	2.45	19.0	88.3	0.77	15.4	0.06	1.04	1.16	11.2
S25	15.4	9.61	8.32	2.68	17.3	1.02	9.37	51.5	0.60	10.0	0.08	1.60	1.43	18.7
S26	15.0	10.9	7.91	5.63	17.5	2.28	42.3	185	3.09	35.5	0.55	17.4	9.96	14.1
S27	1.79	11.2	8.30	2.07	16.8	0.80	8.61	22.8	0.72	15.6	0.08	1.27	1.04	22.3
S28	4.91	10.3	8.66	3.47	17.4	0.72	2.58	26.2	0.47	7.27	0.11	2.54	2.44	15.4
S29	15.6	10.9	7.24	1.67	15.6	1.62	35.7	192	1.14	20.9	0.07	2.71	2.06	15.6
S30	15.1	10.1	6.99	1.63	15.2	1.50	34.5	195	1.12	21.7	0.07	3.21	1.33	15.2
S31	1.53	11.4	6.72	1.36	14.7	1.67	13.0	178	0.78	22.0	0.10	1.82	2.85	7.85
S32	16.1	10.2	8.15	2.62	14.2	2.54	39.4	35.9	1.39	22.9	1.23	1.86	2.30	35.5
S33	1.06	10.2	7.99	2.18	14.9	1.00	5.84	28.0	0.64	5.42	0.06	1.62	1.55	29.7
S34	16.9	10.9	8.09	4.34	15.9	0.91	7.82	27.8	0.52	11.3	0.28	4.47	0.57	31.2
S35	7.52	10.1	7.85	2.20	14.4	0.96	6.85	26.5	0.63	7.18	0.11	2.38	1.27	26.3
S36	1.97	11.4	7.10	2.58	14.9	2.17	24.7	34.5	1.01	17.0	0.21	6.74	1.23	55.0
S37	6.43	6.16	3.88	0.91	5.83	0.75	12.8	19.3	0.28	6.02	0.07	1.67	0.55	9.75
S38	7.56	10.3	8.04	0.30	9.00	4.93	2.81	24.9	0.41	13.3	0.02	0.67	1.33	26.1
S39	12.1	9.93	6.71	1.76	12.7	0.97	8.88	14.7	0.44	13.1	0.14	1.86	0.99	19.8
S40	8.49	10.2	8.06	0.34	8.84	5.52	4.39	28.3	0.42	14.7	0.03	0.85	1.09	29.3
S41	0.94	8.64	7.18	5.03	16.4	0.52	3.58	18.6	0.38	10.3	0.05	2.93	2.42	15.9
S42	1.07	10.1	7.72	4.70	15.1	0.69	8.89	29.1	0.55	9.91	0.13	3.78	1.21	23.7
S43	1.34	10.1	9.91	5.34	15.7	1.42	3.16	23.2	0.41	2.30	0.26	5.51	1.34	71.8
S44	0.78	8.56	6.63	2.75	20.6	3.45	10.4	84.9	0.69	24.9	0.00	8.00	6.55	86.2
S45	1.02	11.0	7.87	1.13	14.3	1.26	3.18	27.4	0.45	18.1	0.10	1.43	1.47	22.0
S46	0.68	6.64	7.38	5.36	26.9	0.46	5.36	32.9	0.93	7.59	0.00	7.07	4.21	4.39

^1^ Sample number.

**Table 3 foods-11-00849-t003:** Relative amounts of Fe^3+^ and Fe^2+^ cations as determined by LCF analysis of Fe K-edge XANES spectra of the *Spirulina* samples.

Sample	Fe^3+^ (%)	Fe^2+^ (%)
S4	92	8
S17	87	13
S22	85	15
S41	82	18
S46	88	12

**Table 4 foods-11-00849-t004:** Compliance with *Spirulina* declared nutrient values (% deviation).

S. No. ^1^	Fe (g/kg)		Mn (mg/kg)		Ca (g/kg)		Zn (mg/kg)		P (g/kg)	
	DV ^2^	MV ^3^ (Deviation (%))	DV	MV (Deviation (%))	DV	MV (Deviation (%))	DV	MV (Deviation (%))	DV	MV (Deviation (%))
S4	1.7	3.48 (+105)	67.0	128 (+91.1)	-	-	-	-	-	-
S5	1.04	0.48 (−53.7)	-	-	-	-	15	14.6 (−2.67)	-	-
S8	0.78	1.39 (+78.2)	-	-	-	-	-	-	-	-
S10	0.07	1.68 (+2453)	-	-	3.33	3.09 (−7.07)	-	-	-	-
S12	1.23	1.38 (+12.2)	67	38.2 (−43.0)	-	-	-	-	9.33	13.9 (+49.0)
S13	0.07	1.79 (+2457)	-	-	3.33	5.39 (+61.9)	-	-	-	-
S14	0.40	0.69 (+72.0)	40	34.9 (−12.8)	-	-	-	-	-	-
S15	0.40	0.69 (+64.5)	40	33.3 (−16.8)	-	-	-	-	-	-
S17	0.30	1.74 (+480)	30	30.9 (+3.00)	1.20	5.34 (+345)	180	34.9 (−80.6)	12.0	13.5 (+12.5)
S23	1.23	1.39 (+13.0)	-	-	-	-	360	14.0 (−96.1)	-	-
S24	1.23	0.77 (−37.6)	-	-	-	-	360	15.4 (−95.7)	-	-
S26	2.30	3.09 (+34.4)	130	185 (+42.3)	-	-	-	-	-	-
S28	0.6	0.47 (−22.3)	-	-	6.67	0.72 (−89.2)	-	-	-	-
S35	0.62	0.63 (+1.29)	-	-	-	-	-	-	-	-
S36	0.62	1.01 (+62.9)	-	-	-	-	-	-	-	-

^1^ Sample number; ^2^ declared Value (DV); ^3^ measured Value (MV): the maximum permissible deviation (% deviation) according to European Commission is is from −20% to +45% [48].

**Table 5 foods-11-00849-t005:** Toxic trace element content in *Spirulina* supplements.

Element	ML ^1^/BMDL_01_ ^2^		Min	Max	Median (IR ^3^)
Cd (μg/kg dwt)	3000		1.22	226	22.6 (13.6–53.9)
Hg (μg/kg dwt)	100		0.84	101	5.57 (3.57–7.63)
Pb (μg/kg dwt)	3000		47.4	1320	355 (167–611)
As (ug/day)	Male (87 kg)	Female (68 kg)			
	26.1–696	20.4–544			
RDI ^4^ 3 g			0.04	8.11	0.82 (0.49–1.10)
RDI ^4^ 10 g			0.15	27.0	2.72 (1.63–3.66)

^1^ Maximum allowed level by European Commission [49]; ^2^ benchmark dose lower confidence limit for cancers of the lung, skin and bladder, and skin lesions determined by The Scientific Panel on Contaminants in the Food Chain (CONTAM) Panel [50]; ^3^ interquartile range; ^4^ recommended daily *Spirulina* intake.

**Table 6 foods-11-00849-t006:** Total amino acid content (mg/day) in *Spirulina* supplements, presented for minimal (3 g) and maximal (10 g) recommended daily intake.

Amino Acid	ADR ^1^ (mg/Day)		Min (mg/Day)		Max (mg/Day)		Median (IR ^2^) (mg/Day)	
	Male (87 kg)	Female (68 kg)	3 g	10 g	3 g	10 g	3 g	10 g
ALA	-	-	62.5	208	148	493	124 (112–132)	414 (374–439)
GLY	-	-	46.9	156	102	341	89.9 (83.1–94.1)	300 (277–314)
VAL	2262	1768	153	509	369	1231	307 (279–325)	1022 (930–1083)
LEU	3393	2652	90.1	300	202	674	169 (160–182)	565 (532–606)
ILE	1740	1360	111	370	257	858	220 (203–231)	733 (677–771)
THR	1305	1020	74.1	247	177	589	149 (137–160)	497 (458–533)
SER	-	-	65.5	218	175	582	140 (127–154)	466 (423–513)
PRO	-	-	52.0	173	105	349	91.6 (85.3–93.9)	305 (284–313)
ASP	-	-	125	415	304	1012	255 (239–270)	850 (796–900)
MET	870	680	34.7	116	74.0	247	61.4 (56.3–66.2)	205 (188–221)
GLU	-	-	218	725	712	2374	498 (455–585)	1661 (1517–1949)
PHE ^3^	2175	1700	63.3	211	121	403	109 (102–112)	362 (341–374)
LYS	2610	2040	90.9	303	377	1256	231 (199–280)	770 (662–934)
HIS	870	680	48.9	163	66.6	222	58.7 (56.3–62.4)	196 (188–208)
TYR ^3^	2175	1700	52.6	175	119	397	104 (96.8–111)	347 (323–371)

^1^ Adult daily requirement (WHO) [61]; ^2^ interquartile range; ^3^ combined recommendations (2175 mg/day total for males and 1700 mg/day total for females).

**Table 7 foods-11-00849-t007:** Fatty acid content (% of total fatty acid content) in *Spirulina* supplements.

Fatty Acid	Min	Max	Median (IR ^1^)
SFA ^2^			
C14:0	0.23	0.63	0.35 (0.31–0.40)
C15:0	<LOD	0.22	<LOD
C16:0	18.4	45.9	41.8 (40.6–42.9)
C17:0	<LOD	0.42	0.33 (0.27–0.36)
C18:0	0.97	25.8	3.90 (1.70–5.46)
C22:0	<LOD	30.2	<LOD
MUFA ^3^			
C14:1	<LOD	1.12	0.33 (0.09–0.41)
C16:1 ω-7	3.07	11.7	6.93 (5.96–7.94)
C17:1	<LOD	0.72	0.35 (0.31–0.42)
C18:1 ω-9c/9t	1.72	9.34	3.78 (3.24–4.38)
PUFA ^4^			
C16:2 ω-4	<LOD	9.38	0.42 (0.36–0.49)
ω-6 PUFA			
C16:2 ω-6	<LOD	9.71	<LOD
LA, C18:2 ω-6c	11.1	31.0	22.0 (20.6–22.7)
GLA, C18:3 ω-6	<LOD	24.1	19.4 (16.7–20.3)
C20:3 ω-6	<LOD	0.82	<LOD (<LOD–0.33)
ω-3 PUFA			
C16:3 ω-3	<LOD	9.58	<LOD
C18:3 ω-3	<LOD	18.4	<LOD

^1^ Interquartile range; ^2^ saturated fatty acids; ^3^ monounsaturated fatty acids; ^4^ polyunsaturated fatty acids; LOD—Limit of Detection.

## Data Availability

Data are contained within the article or Appendix A.

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
