# Peer review of "Nutritional Quality and Safety of the Spirulina Dietary Supplements Sold on the Slovenian Market"

_foods, 2022, doi:10.3390/foods11060849_

Round 1

Reviewer 1 Report

The study compared different spirulina supplements from the markets in terms of their nutritional value. The study is interesting. Some minor comments are required.

  1. L289-296: please transfer this paragraph to the methods section
  2. L321: the PH of what? Gut? It is an in vitro study?
  3. Table 5: please cite the full definition of abbreviations in the footnote of the table.
  4. L404: please define the abbreviations and check them.
  5. 1 and 2 are unclear. Please present the data in tables.
  6. L446: please cite the reference in appropriate form.
  7. L461: please mention them.
  8. Conclusions: What are the author’s recommendations to avoid deviations and adulterations in the commercial products and hazards to the consumer?

Reviewer 2 Report

The manuscript entitled "Nutritional quality and safety characterization of the Spirulina microalgae dietary supplements from the Slovenian market" analyzed the amino acid, fatty acid, and toxic trace element contents of commercial Spirulina supplements sold in Slovenian market in order to determine their safety and nutritional quality. The manuscript is well prepared and the English is good. However, I have some remarks on the manuscript:

  • My major concern is the absence of the statistical analysis in the present study. Only displaying the values of Spirulina analysis is not sufficient. I highly recommend consulting a statistician.
  • The statistical comparison between the collected samples will allow you to recommend the best product(s) in Slovenian market as well as to clarify which is better for consumers to use pure Spirulina products or products with added plant or algae material.
  • Line 44, add reference(s)
  • Line 54, add references
  • In Table 1, in the description of the product contents, the authors sometimes write Spirulina and in other cases write Spirulina platensis. Please make it consistent.

Reviewer 3 Report

The paper deals with a relevant subject and is suitable for publication in this journal. However, some clarifications are needed.

  • Is the number of samples analyzed, representative?
  • Was only one container collected from each sample?
  • What was the study period? At the same time of year?
  • The authors did not clearly report how the analysis of the data obtained was performed.
  • Improve conclusion. It must respond to the objectives of the work. Be careful not to include information that should be added to the discussion of results. 

Reviewer 4 Report

Thank you for the opportunity to review this manuscript.

This study aimed to characterize commercially available Spirulina dietary supplements and their compliance with their given declaration.

This manuscript could be a worthwhile addition to the literature; however, in its present form, this paper is not acceptable and requires some corrections.

Below I provide some suggestions and comments as to how the manuscript could, in my view, be strengthened. I hope the authors will find it useful.

ABSTRACT

  1. This section can be improved. I suggest removing information of lines 22-23 (“For the first time Fe bioavailability in Spirulina is estimated by determining the relative amounts of Fe2+ and Fe3+”) and including the methods used for the dietary supplements characterization.
  2. Line 22: Please include “in Slovenia” or “in the Slovenian market” after “...given declaration”.
  3. Line 26: Please include the recommended amounts in brackets.
  4. There is no mentioning of the results found regarding the compliance with the given declaration. Please include them.

INTRODUCTION

  1. The introduction is logically structured. However, this section can be further developed by deepening its content through the addition of some relevant information.
    1. I suggest authors to include some statistics regarding the estimation of Spirulina consumption in the supplement industry (in Slovenia and worldwide);
    2. I also suggest authors to briefly describe some Spirulina specific characteristics (for example, some morphological and other general characteristics/constituents of this microalgal) to contextualize the reader.
    3. Authors can also further develop the introduction section by adding more information regarding the nutritional properties of Spirulina and its applications on the health context.

The addition of these pieces of information will potentially contextualize the reader and emphasize the importance of the current research.

  1. Lines 60-62: This sentence is referring to the common adulterants found in Spirulina products; however, there are no references for studies. Please provide some literature references to support this statement.
  2. Line 63: “…regular monitoring is
  3. The transition of the third to the last paragraph of the introduction section seems abrupt and disconnected. It is fundamental to clearly provide a justification/knowledge gap of your research and its importance; therefore, I suggest authors to include a justification/knowledge gap that explicits the conduction of this study and its relevance for the scientific literature.

MATERIALS AND METHODS

  1. Line 74 (Samples): In this subsection, authors have described the 46 different Spirulina dietary supplements used in the current study. However, there is no description of the methods used for the selection of the dietary supplements or the selection of the establishments (stores/supermarkets/online stores). Please clarify what were the methods used for their selection.
  2. Table 1 is a bit confusing. I suggest reformulating it by clustering the samples according to their origin or form of presentation (powder, tablets, capsules), for example.
  3. Lines 75-79: I also suggest including in brackets the respective percentages of when referring to the number of the samples tested (i.e.: line 79: n = 34; 73%) throughout the manuscript. Please also check the writing numbers of those lines: numbers up to nine should be written in words, while anything higher can be written in numerals.  
  4. Line 85-87: There are some details that need to be clarified regarding the study samples. How long were the samples kept stored until the analysis? How many samples authors prepared for each Spirulina supplement? Were all the analyses for one supplement performed at the same day? Were all the analyses of all supplements performed at the same day?

RESULTS AND DISCUSSION

  1. The subsection 3.1 (line 235) refers to the results of the elemental composition of the study samples; nonetheless, they are only presented in Table 3. On the other hand, Table 2 is presenting the relative Fe amounts of the samples, which are only mentioned in line 296. Please check the order in which the results have been presented in the manuscript and present them properly.
  2. Line 239-240: Does the recommendation of all Spirulina supplements range from 3-10 g per day – regardless of their form (powder, tablets, capsules)? Please clarify this information in the manuscript.
  3. I suggest authors mention the respective(s) sample(s) in brackets when referring to their results (please check lines 257; line 265; lines 343-348; etc.) to contextualize the reader (authors can take as an example lines 274-277 and lines 281-283).
  4. Line 333: “…more research is needed…”
  5. Table 4 and Table 6: Please use a – (instead of a /) to represent the missing information.
  6. Lines 461-462: It is unclear to me what authors mean with the word ‘sensible’. Please clarify it.

Round 2

Reviewer 2 Report

I have no further comments.

Author Response

There are no further comments.

Reviewer 4 Report

This is an improved version of your manuscript and I support its publication. Thank you for addressing my suggestions/commentaries.

Author Response

There are no further comments.